# BMI1-Mediated Pemetrexed Resistance in Non-Small Cell Lung Cancer Cells Is Associated with Increased SP1 Activation and Cancer Stemness

**DOI:** 10.3390/cancers12082069

**Published:** 2020-07-27

**Authors:** Huan-Ting Shen, Peng-Ju Chien, Shih-Hong Chen, Gwo-Tarng Sheu, Ming-Shiou Jan, Bing-Yen Wang, Wen-Wei Chang

**Affiliations:** 1Institute of Medicine, Chung Shan Medical University, No.110, Sec.1, Jianguo N.Rd., Taichung City 40201, Taiwan; ryenhat@gmail.com (H.-T.S.); gtsheu@csmu.edu.tw (G.-T.S.); msjan@csmu.edu.tw (M.-S.J.); 2Department of Pulmonary Medicine, Taichung Tzu Chi Hospital, Buddhist Tzu Chi Medical Foundation, No. 88, Sec. 1, Fengxing Rd., Tanzi Dist., Taichung City 427, Taiwan; 3Department of Biomedical Sciences, Chung Shan Medical University, No.110, Sec.1, Jianguo N.Rd., Taichung City 40201, Taiwan; chienpengju@gmail.com (P.-J.C.); zx123600@gmail.com (S.-H.C.); 4Immunology Research Center, Chung Shan Medical University, No.110, Sec.1, Jianguo N.Rd., Taichung City 40201, Taiwan; 5Division of Allergy, Immunology and Rheumatology, Department of Internal Medicine, Chung Shan Medical University, No.110, Sec.1, Jianguo N.Rd., Taichung City 40201, Taiwan; 6Division of Thoracic Surgery, Department of Surgery, Changhua Christian Hospital, No. 135, Nanhsiao Street, Changhua County 500, Taiwan; 7School of Medicine, Chung Shan Medical University, No.110, Sec.1, Jianguo N.Rd., Taichung City 40201, Taiwan; 8School of Medicine, College of Medicine, Kaohsiung Medical University, No.100, Shin-Chuan 1st Road, Sanmin Dist., Kaohsiung City 80708, Taiwan; 9Institute of Genomics and Bioinformatics, National Chung Hsing University, No.145, Xingda Rd., South Dist., Taichung City 402, Taiwan; 10Ph.D. Program in Translational Medicine, National Chung Hsing University, No.145, Xingda Rd., South Dist., Taichung City 402, Taiwan; 11Center for General Education, Ming Dao University, No.369, Wen-Hua Rd., Pitou Township, ChangHua County 52345, Taiwan; 12Department of Medical Research, Chung Shan Medical University Hospital, No.110, Sec.1, Jianguo N.Rd., Taichung City 40201, Taiwan

**Keywords:** BMI1, SP1, pemetrexed resistance, non small cell lung cancer, cancer stem cells, epithelial–mesenchymal transition

## Abstract

Lung cancer is the leading cause of cancer death worldwide and the therapeutic strategies include surgery, chemotherapy and radiation therapy. Non-small cell lung cancers (NSCLCs) account for around 85% of cases of lung cancers. Pemetrexed is an antifolate agent that is currently used as the second line chemotherapy drug in the treatment of advanced NSCLC patients with a response rate of 20–40%. The search for any combination therapy to improve the efficacy of pemetrexed is required. The existence of cancer stem cells (CSCs) is considered as the main reason for drug resistance of cancers. In this study, we first found that pemetrexed-resistant NSCLC cells derived from A549 cells displayed higher CSC activity in comparison to the parental cells. The expression of CSC related proteins, such as BMI1 or CD44, and the epithelial–mesenchymal transition (EMT) signature was elevated in pemetrexed-resistant NSCLC cells. We next discovered that the overexpression of BMI1 in A549 cells caused the pemetrexed resistance and inhibition of BMI1 by a small molecule inhibitor, PTC-209, or transducing of BMI1-specific shRNAs suppressed cell growth and the expression of thymidylate synthase (TS) in pemetrexed-resistant A549 cells. We further identified that BMI1 positively regulated SP1 expression and treatment of mithramycin A, a SP1 inhibitor, inhibited cell proliferation, as well as TS expression, of pemetrexed-resistant A549 cells. Furthermore, overexpression of BMI1 in A549 cells also caused the activation of EMT in and the enhancement of CSC activity. Finally, we demonstrated that pretreatment of PTC-209 in mice bearing pemetrexed-resistant A549 tumors sensitized them to pemetrexed treatment and the expression of Ki-67, BMI1, and SP1 expression in tumor tissues was observed to be reduced. In conclusion, BMI1 expression level mediates pemetrexed sensitivity of NSCLC cells and the inhibition of BMI1 will be an effective strategy in NSCLC patients when pemetrexed resistance has developed.

## 1. Introduction

Lung cancer is the leading cause of cancer-related death in the world. Non-small-cell lung cancer (NSCLC) represents 87% of patients with lung cancer, and is clinically divided into two major subtypes: adenocarcinoma and squamous cell carcinoma [1]. Most lung cancer patients present with advanced-stage disease and the current 5-year survival for lung cancer is only 18% [2]. Even in patients with early-stage resectable or locally advanced disease after standard treatment, the lung cancer recurrence rate is up to 90% [3].

Over the past two decades, multiple treatments are under clinical practice for NSCLC patients such as surgery, chemotherapy, targeted therapy, immunotherapy and radiotherapy. However, systemic chemotherapy is still the principal treatment for NSCLC, especially as palliative care for late-stage patients [4]. Platinum-based chemotherapy has been used for standard first-line regimen for advanced NSCLC since 2001 [5]. In addition to platinum, pemetrexed, one of the antifolates, has also been used as a first-line drug to treat NSCLC since 2008 [6]. The interference of folate metabolism by pemetrexed leads to ineffective DNA synthesis and eventually to failure of proliferation in tumor cells. Pemetrexed acts not only with platinum as first-line regimen but also as a single agent for second-line treatment in advanced patients. However, after long-term use of chemotherapy, NSCLC will develop chemoresistance through several mechanisms such as enhanced DNA repair systems, activated epithelial-to-mesenchymal transition (EMT), increased number of cancer stem cells (CSCs), impaired drug uptake, reinforced drug elimination, defective polyglutamylation process, higher expression of oncoprotein thyroid transcription factor-1 (TTF-1) and pemetrexed target enzymes [7]. In NSCLCs, the response rate of pemetrexed treatment is 20–40% [8]. It indicates that resistance to pemetrexed treatment exists among NSCLC patients.

The existence of CSCs is considered as the main reason for the drug resistance of cancers [9,10,11]. The expression of stemness genes has been reported to be associated with chemoresistance in cancers [12,13]. B-cell-specific Moloney leukemia virus insertion site 1 (BMI1), a structural component of the polycomb repressive complex, is functionally associated with the self-renewal of cancer stem cells resulting in the chemo- and radiation resistance of tumors [14]. BMI1 plays an important role in the regulation of gene expression and cellular physiology, including cell cycle control, DNA damage repair, and EMT [15,16,17]. Inhibition of BMI1 can induce apoptosis in tumor cells and elevate the sensitivity of tumor cells to chemotherapy and radiotherapy for tongue and breast cancer [18,19]. BMI1 overexpression was found as an efficient prognostic marker for NSCLC with increased tumor size, poor differentiation, more distant metastasis, and worse survival [20]. Therefore, we designed a study to understand the signaling pathway of BMI1-induced pemetrexed resistance in NSCLC.

In the present study, we firstly observed the upregulation of BMI1 in pemetrexed-resistant NSCLC cells derived from A549 cells. The overexpression of BMI1 in A549 cells caused pemetrexed resistance and knockdown of BMI1 in resistant cells, sensitized them to pemetrexed treatment, and vice versa. We also demonstrated that BMI1 could induce Specificity protein 1 (SP1) expression and lead to the upregulation of thymidylate synthase (TS, gene name as TYMS), a key enzyme for pemetrexed resistance. In addition, we found that BMI1 induced the EMT process and enhanced cancer stemness when overexpressed in A549 cells. We also provided the evidence that pre-treatment with PTC-209, a BMI1 inhibitor, could sensitize NSCLC tumors derived from pemetrexed-resistant cells to pemetrexed treatment.

## 2. Results

### 2.1. CSC Activity and EMT Signature Is Upregulated in Pemetrexed-Resistant NSCLC Cells

We have established the pemetrexed-resistant NSCLC cells from A549 cells, called A400 cells [21]. Given the important role of CSC in drug resistance [9], we first compared the CSC activity between A549 and A400 cells. By tumorsphere cultivation, the CSC activity was greater in A400 cells than that of A549 cells (Figure 1A). BMI1 [14], CD44 [22], or Notch1 [23] has been reported as the important positive regulator in the self-renewal capability of lung CSCs. By western blot analysis, the protein expressions of BMI1, CD44, or activated Notch1 (notch intracellular domain, NICD), were upregulated in A400 cells in comparison to parental A549 cells (Figure 1B). Aldehyde dehydrogenase (ALDH) isoforms, such as ALDH1A1 or ALDH1A2, have been reported as CSC markers in several cancer types [24,25]. However, we observed that the expression of ALDH1A1 or ALDH1A2 was reduced in pemetrexed A400 cells when compared to the parental A549 cells (Figure 1C). It was consistent with a previous report from Okudela K et al., in which it was described that ALDH1A1 might function as a tumor suppressor in NSCLC [26]. In addition, we also observed the elevated EMT signature in pemetrexed-resistant A400 cells by the downregulation of E-cadherin, the epithelial marker, and the upregulation of N-cadherin, the mesenhymal marker, and Snail1/ZEB1, the transcription factors involved in the EMT process (Figure 1D). These data suggest that the pemetrexed resistance in NSCLC cells may be associated with the increased CSC activity and the elevated EMT signature.

### 2.2. The Expression Level of BMI1/Sp1/Thymidylate Synthase Is Correlated with Pemetrexed Sensitivity in NSCLC Cells

We also took another NSCLC cell line, H1355, to compare the pemetrexed sensitivity and the results displayed that A549 were the most sensitive NSCLC cells followed by H1355 and A400 cells (Figure 2A). It has been reported that the upregulation of thymidylate synthase (TS) is one of the reasons for pemetrexed resistance [27]. Overexpression of BMI1 is also found in cancer cells with resistance to chemotherapy agents [14]. We next compared the expression of BMI1, SP1, or TS in A549, A400, or H1355 NSCLC cells by western blot. All these three protein’s expressions in A400 or H1355 cells were higher than those of A549 cells (Figure 2B). Here, we hypothesize that the upregulation of BMI1/SP1 may lead to TS overexpression and pemetrexed resistance in NSCLC cells.

### 2.3. Manipulation of BMI1 Expression Level in NSCLC Cells Changes the Pemetrexed Sensitivity

We next examined if overexpression of BMI1 in A549 cells could induce pemetrexed resistance. From Figure 3A, the overexpression of BMI1 in A549 cells induced pemetrexed resistance in comparison to control cells (Figure 3A). We also found that Sp1 expression was upregulated in BMI1-overexpressed A549 cells (Figure 3B). To further investigate the effects of BMI1 inhibition in pemetrexed-resistant A400 cells, the knockdown of BMI1 in A400 cells was performed by lentiviral delivery of specific shRNAs (Figure 3C). The decreased cell growth of A400 cells was observed after knockdown of BMI1 with or without pemetrexed treatment (Figure 3D). Given that the sensitivity of pemetrexed in NSCLC cells was thought to be associated with the level of TS expression [28], we next checked the expression of TS in NSCLC cells after inhibition of BMI1 protein expression or its bioactivity. The treatment of a BMI1 inhibitor, PTC-209, caused the down-regulation of TS in both A400 and H1355 cells (Figure 4A). We also found that the treatment of PTC-209 in pemetrexed A400 cells caused the cell cycle arrest at the G1 phase in a dose-dependent manner (Figure 4B). The knockdown of BMI1 by specific shRNAs also suppressed TS expression in both A400 and H1355 cells (Figure 4C). From these results, it suggests that BMI1 expression level determines pemetrexed sensitivity in NSCLC cells.

### 2.4. BMI1 Is the Upstream Signaling Molecule of Sp1 Protein Expression in Pemetrexed-Resistant NSCLC Cells

It has been reported that SP1 could trans-activate TS expression in MCF7 breast cancer cells [29]. We next examined the role of SP1 in pemetrexed NSCLC cells. The protein expression of Sp1 was more upregulated in A400 cells than that of parental A549 cells (Figure 5A). Overexpression of BMI1 in A549 cells upregulated both SP1 and TS expression (Figure 5B). The SP1 inhibitor, mithramycin A (MitA), was used to treat pemetrexed-resistant A400 cells and results showed that MitA dose-dependently suppressed the proliferation of A400 cells with or without pemetrexed treatment (Figure 5C). MitA treatment in A400 or H1355 cells dose-dependently inhibited TS expression (Figure 5D). Using PTC-209 to inhibit BMI1 activation in A400 cells, we found that PTC-209 also inhibited SP1 protein expression (Figure 5E). We further found that there was a positive correlation between BMI1 and SP1 mRNA expression in the Cancer Genome Atlas (TCGA) database using the Genen Expression Profiling Interactive Analysis webtool (GEPIA, http://gepia.cancer-pku.cn/index.html) (R = 0.64, *p* = 0, Figure 5F). It indicates that BMI1 is an upstream signaling molecule of Sp1 or TS expression in pemetrexed NSCLC cells.

### 2.5. BMI1 Overexpression Activates EMT and Enhances Cancer Stemness in NSCLC Cells

It has been suggested that cancer chemoresistance could be induced by the EMT program or be associated with cancer stemness. From the results in Figure 1C, an increased EMT process was found in pemetrexed A400 cells in comparison to parental A549 cells. We firstly examined if knockdown of BMI1 influences the EMT process in A400 cells and results revealed that the expression of E-cadherin was increased while the transcriptional repressors of E-cadherin, such as Snail1 or ZEB1, were decreased after knockdown of BMI1 (Figure 6A). We next examined the changes in the EMT process in A549 cells after overexpression of BMI1. By western blot analysis, the downregulation of E-cadherin and the upregulation of N-cadherin was simultaneously observed in BMI1-overexpressed A549 cells (Figure 6B), which suggests the activation of the EMT program by BMI1. The upregulation of Snail or ZEB1 was also observed in BMI1-overexpressed A549 cells (Figure 6B). In addition, the cell migration capability of A549 cells was also enhanced by BMI1 overexpression (Figure 6C). Using tumorsphere cultivation, the overexpression of BMI1 in A549 cells increased CSC activity (Figure 6D). Treatment of PTC-209 in pemetrexed-resistant A400 cells inhibited tumorspere number in a dose dependent manner (Figure 6E). These data suggest that BMI1 overexpression in NSCLC cells activates the EMT program and cancer stemness, which may lead to pemetrexed resistance.

### 2.6. PTC-209 Sensitizes NSCLC Tumors toward Pemetrexed Treatment In Vivo

We finally examined the in vivo therapeutic potential of PTC-209 for the treatment of NSCLC tumors with pemetrexed resistance. The pemetrexed-resistant A400 NSCLC cells were subcutaneously inoculated into the back skin of NOD/SCID mice and divided into two groups when the tumor volume reached 50 mm^3^, with group 1 as the vehicle control and group 2 as the PTC-209-treated group, with 20 mg/kg for 5 days/week for a total of 3 weeks. After one week of the last PTC-209 treatment, all the mice received 100 mg/kg pemetrexed for two times with an interval of 7 days as outlined in our previous report [21]. The mice were then sacrificed at Day 7 after the last pemetrexed treatment and the tumor weight was measured. The weight of tumors was significantly lower in the PTC-209-pretreated group (Figure 7A). The percentage of tumor cells with BMI1 or Ki67, the marker for cell proliferation, and the mean staining intensity of SP1 expression was significantly reduced in the PTC-209-pretreated group (Figure 7B). These results suggest that PTC-209 is a potential combination agent for NSCLCs treatment when pemetrexed resistance has developed.

## 3. Discussion

The clinical use of pemetrexed was started in treating mesothelioma in 2003 [30]. As the second-line treatment for NSCLC, pemetrexed offers clinically equivalent efficacy outcomes but fewer side effects compared with docetaxel [31]. Based on better tolerability and more convenient administration, platinum with pemetrexed was suggested as standard first-line regimen for advanced NSCLC in 2008 [32]. Maintenance therapy with pemetrexed in patients who received the preceding induction therapy with platinum-pemetrexed revealed improved progression-free and overall survival in advanced NSCLC [33].

From the perspective of patients, not only the drug efficacy but also the quality of life is equally important for cancer patients. The properties of pemetrexed, such as good efficacy, convenient administration and fewer side effects, make it a widely acceptable drug among lung cancer patients. Reversing the pemetrexed resistance means patients can afford lower medical expenses and have a reasonable quality of life. Patients who developed drug resistance to pemetrexed may still have other treatment options, including second-line chemotherapy, tyrosine kinase inhibitors [34], or immunotherapy [35]. However, the above mentioned treatments are either too expensive or associated with lots of side effects. Using the combination of other drugs to make NSCLC patients regain their sensitivity to pemetrexed will have a positive impact on the financial burden of patients.

The biochemical changes in tumor cells with chemoresistance include oncogene or oncoprotein regulations, enhanced DNA repair systems, activated EMT, and an increased number of CSCs [7,36]. In addition, the altered pharmacodynamics and pharmacogenetics of cancer cells also lead to chemoresistance. For example, the expression level of ATP-binding cassette transporters or multidrug resistance-associated proteins could facilitate the removal of anti-cancer drugs from cancer cells [37,38,39]. In addition, the polymorphisms of the enzymes involved in xenobiotic metabolisms, such as the members of the cytochrome P450 family [40], glutathione S-transferases, or diphosphate glucuronosyltransferases, have been frequently mentioned among the influences of pharmacokinetics of anti-cancer drugs [41,42]. Our study focuses on the oncogene BMI1 because it plays an important role in the development of tumor metastasis, stemness maintenance, and chemo- and radiotherapy resistance. Siddique et al. previously demonstrated that BMI1 was overexpressed in docetaxel-resistant prostate cancer cells [43]. Knockdown of BMI1 in docetaxel resistant prostate cancer cells inhibited the expression of cyclin D1 or BCL2, the downstream targets of the Wnt/β–catenin pathway. The overexpression of BMI1 in prostate cancer cells also enhanced the transcriptional activity of TCF [43]. The induction of the noncanonical Wnt pathway by BMI1 was also observed in melanoma cells that were resistant to the treatment of BRAF inhibitor by the upregulation of the Wnt5a and ROR2 receptors [44]. The overexpression of BMI1 in melanoma cells induced the invasive signature with concomitant nuclear accumulation of β-catenin [44]. Here, we described another pathway for BMI1-induced chemoresistance in NSCLCs through the upregulation of SP1 activity followed by the upregulation of TS, the elevated CSC activity, and turning on the EMT process. Using GEPIA to analyze the TCGA database, the mRNA expression between *BMI1* and *TYMS* or *SP1* and *TYMS* displayed a significantly positive correlation among lung adenocarcinoma patients (Appendix A, R = 0.42, *p* = 0 for *BMI1*/*TYMS*; R = 0.35, *p* = 3.6 × 10^−15^ for *SP1*/*TYMS*). Such observations from clinical data in the TCGA database strongly support our findings.

SP1 is one of the transcription factors and belongs to the Sp/Krüppel like factor (KLF) family that plays multiple roles in maintaining cellular homeostasis and critical factors in diseases including cancer [45]. In lung cancer, SP1 expression is correlated with the tumor progression [46,47,48]. Wang HB, et al. found the core promoter regions of BMI1, and identified SP1 as an important transcription factor directly binding to the BMI1 promoter region in nasopharyngeal cancer [49]. However, our findings demonstrated that BMI1 could induce SP1 expression but inhibition of SP1 didn’t suppress the BMI1 activity in pemetrexed-resistant A400 cells (Appendix A). It indicates that SP1 is a downstream molecule of BMI1 activation in pemetrexed NSCLC cells. We found that the expression level of BMI1 was consistent with that of TS (Figure 2B), which has been reported to be positively correlated to the pemetexed-resistance [50,51]. We observed that the treatment of PTC-209 in pemetrexed A400 cells enhanced the therapeutic effect of pemetrexed on A400 xenograft tumors in vivo (Figure 7A). Yong et al. have demonstrated that the treatment of PTC-209 for one month in a mouse lung cancer model of C/EBPα (CCAAT/enhancer-binding protein α) deletion reduced the tumor burden by up to 70% [52]. PTC-596, an orally active BMI1 inhibitor, has been used for phase I evaluation in patients with advanced solid tumors (https://clinicaltrials.gov/ct2/show/NCT02404480). We also observed the in vitro growth inhibition of Mithramycin A to pemetrexed A400 cells (Figure 5C). Mithramycin A has been used for phase II evaluation in chest cancers including lung cancers (https://clinicaltrials.gov/ct2/show/NCT01624090). Our results indicate that pretreatment of PTC-209 could re-sensitize pemetrexed-resistant A400 tumors to pemetrexed (Figure 7). It suggests the potentially clinical use of the inhibitors of BMI1 in NSCLC patients with pemetrexed resistance.

## 4. Materials and Methods

### 4.1. Chemicals and Reagents

PTC-209, the BMI1 inhibitor, and mithramycin A, the SP1 inhibitor, were purchased from Cayman Chemicals (Ann Arbor, MI, USA) and dissolved in dimethyl sulfoxide (DMSO, Sigma-Aldrich, St. Louis, MO, USA) at 2 mM or 18 mM, respectively, and stored at −20 °C. Pemetrexed disodium heptahydrate was purchased from Eli Lilly and Company (Indianapolis, IN, USA) and dissolved in sterile ddH_2_O at 100 mg/mL and stored at −20 °C. Puromycin and blasticidin S were purchased from TOKU-E (Bellingham, WA, USA) and dissolved in PBS at 20 mg/mL and stored at −20 °C.

### 4.2. Cell Culture

A549 or H1355 cells were purchased from American Type Culture Collection (ATCC, Manassas, VA, USA) and were maintained in Dulbecco’s Modified Eagle Medium (DMEM, Gibco^TM^, Waltham, MA, USA) containing 10% fetal bovine serum (FBS, HyClone Laboratories Inc., South Logan, UT, USA) supplied with 1 mM glutamine (Gibco), 1 mM sodium pyruvate (Gibco), 1X Non-Essential Amino Acids (Gibco), and 100 U/mL penicillin/streptomycin (Gibco) at 37 °C, 5% CO_2_ incubator. A400 cells were derived from A549 cells with pemetrexed resistance, which were established as outlined in the previous report [21] and maintained in DMEM/10% FBS as used for A549 cells, except for the addition of 400 ng/mL pemetrexed.

### 4.3. Overexpression or Knockdown of BMI1 by Lentivirus Transduction

The BMI1 gene was first amplified from A400 cDNA and cloned into p3XFlag-CMV-13 vector (Sigma-Aldrich) with the following primers: 5′-CGCGGTACCATGCATCGAACAACGAGAATC-3′/5′-AGCGGATCCACCAGAAGAAGTTGCTGATGAC-3′ after KpnI and BamHI digestion. The flag-tagged BMI1 gene was further amplified and subcloned into pLAS5w.Pbsd-L-tRFP-C lentiviral vector (the National RNAi Core Facility at Academia Sinica, Taipei, Taiwan) with the following primers: 5′-CGCGCTAGCATGCATCGAACAACGAGAATC-3′/5′-GCGTCTAGACTACTTGTCATCGTCATCCTTG-3′ after NheI and XbaI digestion. BMI1-specific shRNA plasmids (TRCN0000020156 and TRCN0000229416) were obtained from the National RNAi Core Facility at Academia Sinica. The package of lentivirus was performed in 293T cells according to the protocol of our previous report [53]. The transduction of lentivirus was performed by the addition of 8 µg/mL polybrene (Sigma). For selection of successfully transduced tRFP/BMI1-tRFP or shRNAs, the cells were treated with 20 µg/mL blasticidin S or 2 µg/mL puromycin, respectively.

### 4.4. Western Blot

Total cellular proteins were obtained by lyzed with RIPA buffer (Genetex Inc., Irvine, CA, USA) and the protein concentration was quantified with the Pierce™ BCA Protein Assay kit (Thermo Fisher Scientific, Waltham, MA, USA). An amount of 25 µg of total cellular proteins was separated by SDS-PAGE and transferred onto polyvinylidene fluoride membrane (Pall Corporation, Washington, NY, USA). After blocking with 1% skim milk/TBS-T (20 mM Tris, 150 mM NaCl, 0.1% Tween-20) at room temperature (RT) for 1 h, the membrane was incubated with primary antibody at 4 °C overnight followed by horseradish peroxidase conjugated secondary antibody at RT for 1 h. The signal was then developed by Pierce™ ECL Western Blotting Substrate (Thermo Fisher) and captured by the FUSION Solo S imaging system (Marne-la-Vallée, France). The antibodies used in this study were listed in Appendix A.

### 4.5. Tumorsphere Cultivation

Cells were harvested by trypsin/EDTA and were suspended at 2000 cells/2 mL/well of ultralow attachment 6-well-plate (Greiner Bio-One GmbH, Kremsmünster, Austria) at 37 °C/5% CO_2_ incubator in tumorsphere media at DMEM/F12 (Gibco) containing 0.4 mg/mL bovine serum albumin (Gibco), 0.01% methylcellulose (Sigma-Aldrich), 20 ng/mL epidermal growth factor (PeproTech, Inc., Rocky Hill, NJ, USA), 20 ng/mL basic fibroblast growth factor (PeproTech), 5 µg/mL insulin (Sigma-Aldrich), 4 µg/mL heparin (Sigma-Aldrich), 1 µM hydrocortisone (Sigma-Aldrich). Wells were added with 300 µL of fresh tumorsphere media every 4 days and the formed primary tumorspheres were counted at Day 21 under inverted light microscopy (Motic Incorporation Ltd., Hong Kong) and collected with 100 µm cell strainer (BD Biosciences, Franklin Lakes, NJ, USA). After dissociation with HyQTase (Hyclone), 1000 cells from primary tumorspheres were used for secondary tumorsphere cultivation as the condition used for primary tumorspheres.

### 4.6. Wound Healing Assay

Cell migration ability was determined by wound healing assay with a 2 well silicone insert (ibidi GmbH, Gräfelfing, Germany). Briefly, the silicone insert was placed in a 3.5 cm cell culture dish and loaded with 3 × 10^4^ cells for attachment overnight. After removing silicone insert, the dish was filled with complete culture medium and was pictured under an inverted light microscopy (Motic) before being put back into a 37 °C/CO_2_ incubator. The dish was pictured at 3, 6, 12, and 24 h and the cell migration area was analyzed by calculating the relative percentage of area without cell coverage using ImageJ software (Version 1.53a, National Institutes of Health, Bethesda, MD, USA).

### 4.7. NSCLC Xenograftment in NOD/SCID Mice

The NSCLC xenograftment experiments in NOD/SCID mice were approved by the Institutional Animal Care and Use Committee in Chung Shan Medical University (Taichung, Taiwan) with the approval No. 2017. A400 cells (2.5 × 10^6^) were suspended in 100 µL of 2.5 mg/mL matrigel (BD Biosciences) and were subcutaneously injected at the right site of back part for tumor growth. As the tumor volume reached 50 mm^3^ by a calculation formula of 1/2 (length × width^2^) [54], the mice were divided into two groups; vehicle control (20% (w/w) Cremophor EL in 0.9% NaCl) and PTC-209/pemetrexed treatment. For the PTC-209/pemetrexed group, the tumor bearing mice firstly received PTC-209 at 20 mg/kg/day for 5 days a week for a total of 3 weeks and pemetrexed treatment was performed at 100 mg/kg once per week for a total of 2 weeks. The mice were sacrificed at Day 95 after injection of A400 cells and the xenografted tumors were harvested, weighted, and fixed with 3.7% formaldehyde (Sigma-Aldrich) for immunohistochemistry analysis.

### 4.8. Immunohistochemistrical Analysis

The formaldehyde fixed tumor tissues were embedded into paraffin and sliced into 4 µm sections. After deparaffinization steps, the slides were incubated with primary antibodies at 4 °C overnight followed by HRP-conjugated secondary antibodies at room temperature for 1 h. After development with 3,3′-diaminobenzidine substrate (VECTOR Laboratories, Inc., Burlingame, CA, USA), counter staining was performed on the slides with hematoxylin (Sigma-Aldrich) followed by scanning with the TissueFAXS PLUS system and quantification with TissueFAXS Imaging Software (version 7.0, TissueGnostics GmbH, Vienna, Austria).

### 4.9. Cell Cycle Analysis

A400 (1.5 × 10^5^) cells were seeded into 3.5 cm cell culture dishes and treated with PTC-209 or 0.1% DMSO for 24 h followed by harvesting with trypsin/EDTA. After counting the cell number with trypan blue, 1 × 10^5^ cells were used for fixation with 70% EtOH/PBS solution at 4 °C overnight. The fixed cells were then stained with 20 µg/mL PI in 0.1% Triton X-100/PBS at room temperature for 30 min. After being resuspended in 0.2 mg/mL RNase/PBS solution, the fluorescence signals of PI were collected by flow cytometry (BD FACSCalibur™, BD Biosciences, Franklin Lakes, NJ, USA) and the data were analyzed by FlowJo software (version 10, BD Biosciences).

### 4.10. Statistical Analysis

Quantitative data were presented as the mean ± SD. The comparisons between the two groups were analyzed with an unpaired *t*-test. The comparisons among multiple groups (more than two) were analyzed with a repeated measure ANOVA followed by a Tukey–Kramer post hoc test to identify differences among specific groups. A p value of less than 0.05 was considered statistically significant.

## 5. Conclusions

Our data revealed that BMI1 was upregulated in pemetrexed-resistant NSCLC cells. The overexpression of BMI1 in pemetrexed-sensitive NSCLC cells also reduced the efficacy of pemetrexed. The involvement of BMI1 in pemetrexed resistance may result from three possible pathways including (1) the upregulation of SP1 followed by increasing TS expression; (2) the induction of the EMT process by upregulating Snail1 or ZEB1 expression; (3) the elevation of CSC activity (Figure 8). It also suggests that the inhibition of BMI1 may be a potential strategy to overcome pemetrexed resistance in NSCLC patients.

## Figures and Tables

**Figure 1 cancers-12-02069-f001:**
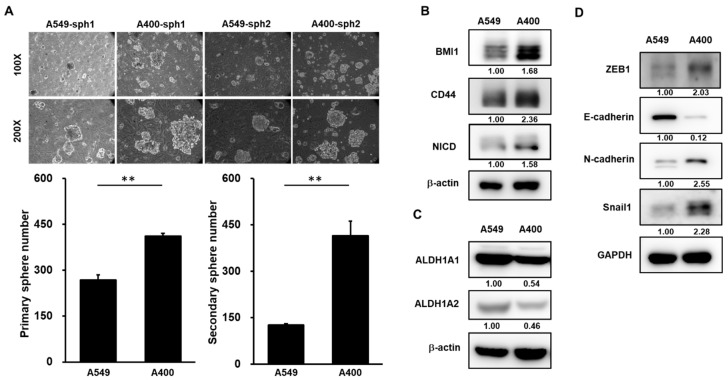
Cancer stem cell (CSC) activity and epithelial-to-mesenchymal transition (EMT) signature is upregulated in pemetrexed-resistant non-small-cell lung cancer (NSCLC) cells. (**A**) The CSC activity in A549 cells or their derived pemetrexed-resistant A400 cells was determined by tumorsphere cultivation. Primary tumorspheres (sph1) were counted at Day 21 by an inverted light microscopy after cultivation. Single cells of primary tumorspheres were obtained by HyQTase dissociation and used for secondary tumorsphere formation (sph2). ** *p* < 0.01. (**B**,**C**) The total proteins were collected from A549 or A400 cells and western blotting was performed to determine the expression of cancer stemness factors (**B**), aldehyde dehydrogenase (ALDH) isoforms (**C**), or EMT-related proteins (**D**). All the experiments were done two times and data from one experiment were presented.

**Figure 2 cancers-12-02069-f002:**
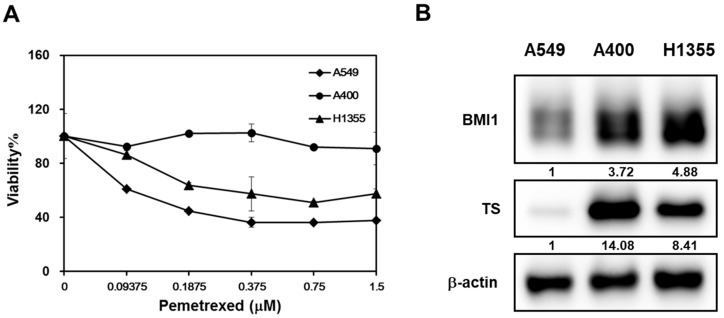
The expression level of B-cell-specific Moloney leukemia virus insertion site 1 (BMI1)/Specificity protein 1 (SP1)/thymidylate synthase (TS) is correlated with pemetrexed sensitivity in NSCLC cells. (**A**) Three NSCLC cells (A549, A400, or H1355) were seeded into a 96-well-plate at 1000 cells/well and treated with the indicated concentration of pemetrexed. The cell viability was determined by 3-(4,5-dimethylthiazol-2-yl)-2,5-diphenyltetrazolium bromide (MTT) reagent at 96 h after treatment. (**B**) The total proteins were harvested from three NSCLC cells and the expression of BMI1 or TS was determined by western blot. All the experiments were done three times and data from one experiment were presented.

**Figure 3 cancers-12-02069-f003:**
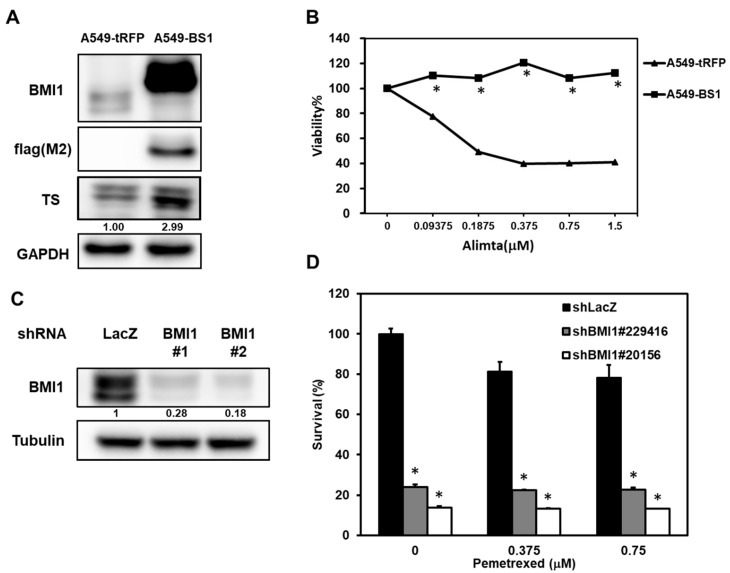
Manipulating BMI1 expression in NSCLC cells influences the pemetrexed sensitivity. (**A**,**B**) A549 cells were transduced with lentiviruses of tRPF control (A549-tRFP) or flag-tagged-BMI1 (A549-BS1) and selected with 20 µg/mL blasticidin S for 96 h. The expressions of flag tagged-BMI1 and TS in A549-tRFP or A549-BS1 were examined by western blot (**A**). The pemetrexed sensitivity of A549-tRFP or A549-BS1 cells were determined by MTT assay (**B**). * *p* < 0.05. (**C**,**D**) A400 cells were transduced with lentiviruses carrying sh-LacZ or sh-BMI1 (#1 or #2) and selected with 2 µg/mL puromycin for 48 h. The survived cells were harvested and BMI1 expression was examined by western blot (**C**) or pemetrexed sensitivity was examined by MTT assay (**D**). * *p* < 0.05. All the experiments were done three times and data from one experiment were presented.

**Figure 4 cancers-12-02069-f004:**
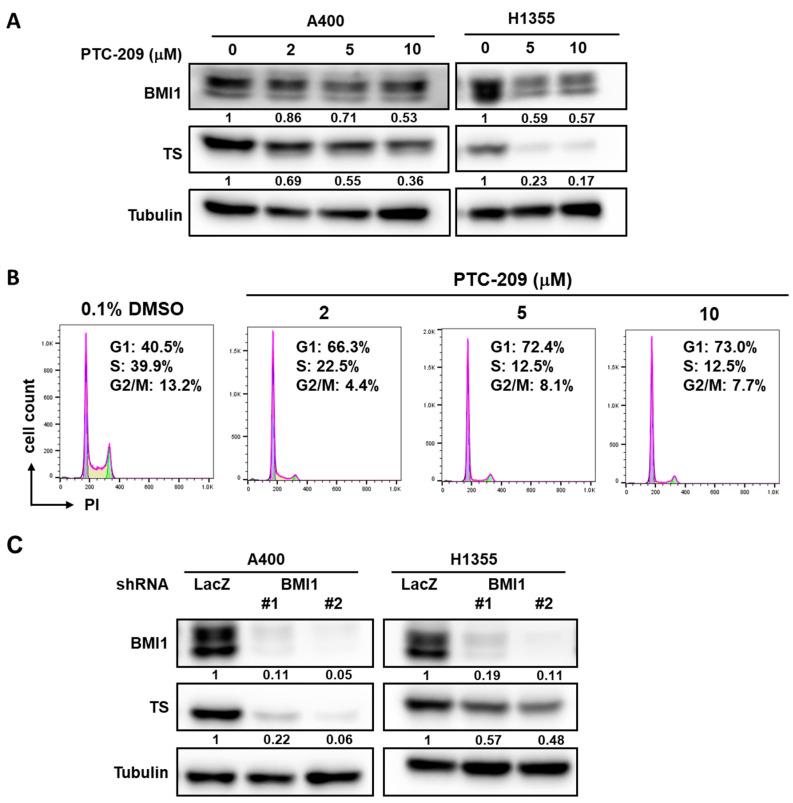
Inhibition of BMI1 in NSCLC cells down-regulates TS expression. (**A**) BMI1 inhibitor, PTC-209, was used to treat A400 or H1355 NSCLC cells at the indicted concentration for 48 h and the total proteins were harvested for determination of BMI1 or TS expression by western blot. (**B**) A400 cells were treated with the indicated concentration of PTC-209 for 24 h and harvested for cell cycle analysis using propidium iodide (PI) staining and the fluorescence was detected by a flow cytometry. As a vehicle control, 0.1% dimethyl sulfoxide (DMSO) was used. The percentage of cells at each cell cycle phase was quantified by FlowJo software. (**C**) A400 or H1355 NSCLC cells were transduced with lentiviruses carrying sh-LacZ or sh-BMI1 (#1 or #2) for 48 h and the protein expression of BMI1 or TS was determined by western blot. All the experiments were done two times and data from one experiment were presented.

**Figure 5 cancers-12-02069-f005:**
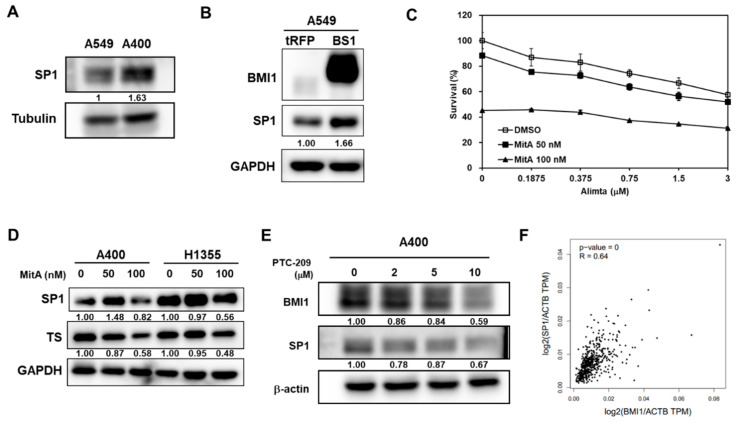
BMI1 is an upstream signaling molecule of SP1 protein expression in pemetrexed-resistant NSCLC cells. (**A**) SP1 expression in A549 or A400 cells was determined by western blot. (**B**) BMI1 and SP1 expression in tRFP- or BMI1-overexpressed A549 (BS1) were determined by western blot. (**C**) Mithramycin A (MitA), the SP1 inhibitor, was used to treat A400 cells at 50 or 100 nM with co-treatment of pemetrexed at the indicated concentration. DMSO was used as a negative control. The cell survival was determined by MTT assay. (**D**) A400 or H1355 cells were treated with the indicated concentration of MitA for 48 h. The protein expression of SP1 or BMI1 was then determined by western blot. (**E**) A400 cells were treated with the indicated concentration of PTC-209 for 48 h and the protein expressions of BMI1 or SP1 were determined by western blot. (**F**) The correlation between mRNA of SP1 and BMI1 among lung adenocarcinoma patients of the Cancer Genome Atlas (TCGA) database was analyzed by the Genen Expression Profiling Interactive Analysis webtool (GEPIA) website. The experiments of (**A**) to (**E**) were done three times and data from one experiment were presented.

**Figure 6 cancers-12-02069-f006:**
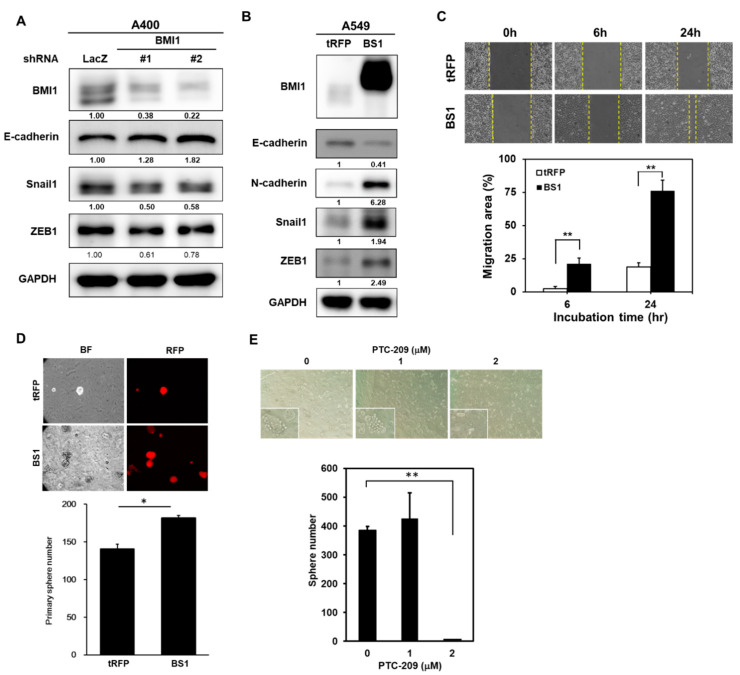
Overexpression of BMI1 induces EMT in A549 NSCLC cells. (**A**) A400 cells were transduced with lentiviruses carrying shLacZ, shBMI1#1, or shBMI1#2, respectively, for 72 h and the total cellular proteins were collected for western blot analysis of BMI1 or the EMT-related molecules including E-cadherin/Snail1/ZEB1. GAPDH was used as a protein loading control. (**B**–**D**) A549 cells were transduced with lentiviruses of tRPF control (A549-tRFP) or flag-tagged-BMI1 (A549-BS1) and selected with 20 µg/mL blasticidin S for 96 h. The protein expressions of BMI1 and the EMT markers including E-cadherin/N-cadherin/Snail1/ZEB1 were determined by western blot (**B**). Cell migration abilities of A549-tRFP or A549-BS1 cells were analyzed by wound healing assay (**C**). ** *p* < 0.01. The CSC activity of A549-tRFP or A549-BS1 cells were assessed by tumorsphere cultivation (**D**). The red fluorescent protein (RFP) signals (upper panel) and tumorsphere numbers (lower panel) were captured and counted with an inverted fluorescence microscopy at Day 14. * *p* < 0.05. (**E**) Tumorsphere cultivation was performed on A400 cells in the presence of 1 or 2 µM PTC-209 and the formed tumorspheres were pictured (upper panel) and counted (lower panel) at Day 14. ** *p* < 0.01 as compared to DMSO control. All the experiments were done two times and data from one experiment were presented.

**Figure 7 cancers-12-02069-f007:**
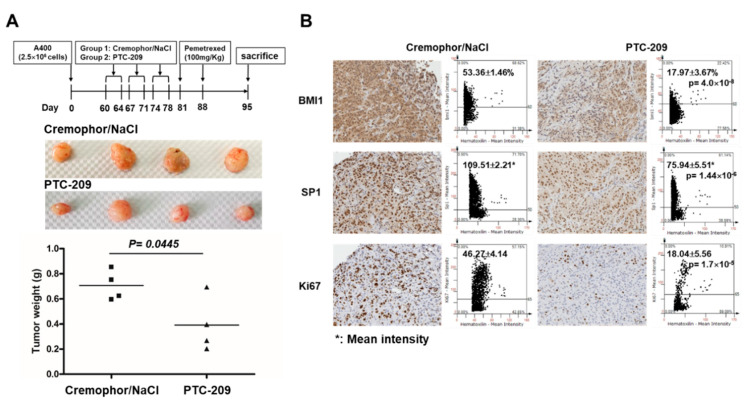
PTC-209 sensitizes NSCLC tumors toward pemetrexed treatment in vivo. Pemetrexed-resistant A400 cells were subcutaneously inoculated into the back skins of NOD/SCID mice as 2.5 × 10^6^ cells/mouse (n = 4). After tumor volume reached to 50 mm^3^, the mice were divided into two groups: group 1: vehicle (Cremophor/NaCl) or group 2: PTC-209 treatment with intraperitoneal injection at a dose of 20 mg/kg for 5 days/week, the treatments were done for a total of 3 weeks. After 7 days from last shot of vehicle or PTC-209, all the mice were treated with pemetrexed at a dose of 100 mg/kg for two times with an interval of 7 days. The mice were then sacrificed at Day 7 after the last treatment of pemetrexed. (**A**) Tumors were harvested and the weight was compared between vehicle pre-treated control group and PTC-209 pre-treated group. (**B**) Paraffin embedded tumor tissues were sliced into 4 µm sections and immunohistochemistry analysis was performed to evaluate the protein expression of BMI1, SP1, and Ki67. The images were quantified by TissueFAX software to measure the positive percentages or mean staining intensity.

**Figure 8 cancers-12-02069-f008:**
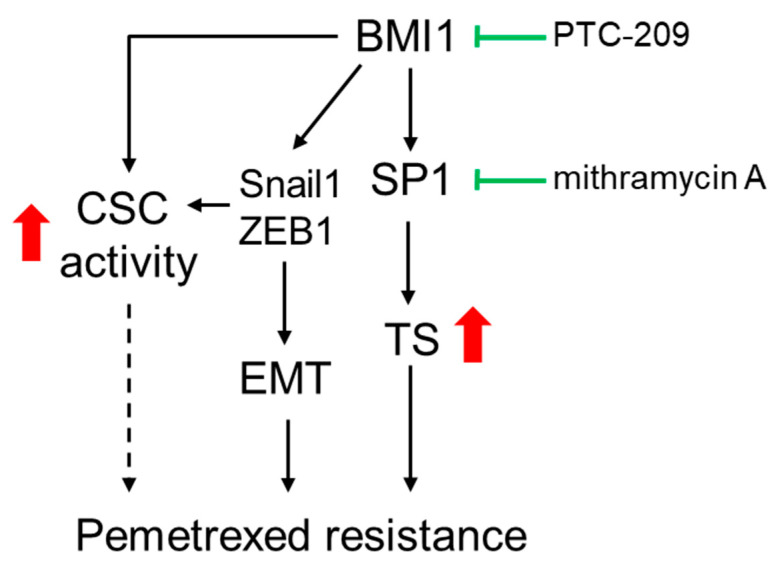
The proposed model for BMI1 involved pemetrexted resistance in NSCLC. The role of BMI1 in pemetrexed resistance could result from three ways including (1) the upregulation of SP1 followed by increased TS expression; (2) The enhancement of the EMT process by upregulating Snail1 or ZEB1 expression; (3) The elevation of CSC activity. Inhibition of BMI1 or SP1 activity by PTC-209 or mithramycin A may serve as the potential agents for overcoming the pemetrexed resistance in NSCLC patients.

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
