# Peer review of "BMI1-Mediated Pemetrexed Resistance in Non-Small Cell Lung Cancer Cells Is Associated with Increased SP1 Activation and Cancer Stemness"

_cancers, 2020, doi:10.3390/cancers12082069_

Round 1
Reviewer 1 Report
The manuscript by Shen et al is an interesting study, however, there are some points that need to be improved:
- It will be interesting to determine whether treatment with pemetrexed induces the expression of BMI1 or contrary it selects those tumor A549 cells that express increased levels of BMI1, previous to pemetrexed treatment. Therefore, authors can quantify BMI1, TS expression +/- pemetrexed, 1.5 uM, treatment at short time and check possible induction of the expression of these proteins.
- Figure 5B. Panel showing TS expression is missing.
- NSCLC xenografts experiments in NOD/SCID mice do not have enough controls. Authors demonstrate that PTC-209 treatment of A400-tumors enhanced the therapeutic effect of pemetrexed. But, as understood in the section of Materials and Methods, only the group of tumors treated with PTC-209 are also treated with pemetrexed, but not the vehicle control group. To ascertain the role of increased BMI1 in the pemetrexed treatment, A400-control tumors (which supposedly express high levels of BMI1) must also be treated with pemetrexed, and then compared with A400-PTC-209/pemetrexed tumors (expressing disminished levels of BMI1).
- Measure of levels of expression of BMI1, SP1 and TS in all type of A-400 tumors (with/without PTC-209, pemetrexed treatments) is needed.
- In the Discussion section, if authors suggest the potential clinical use of the inhibitors of BMI1 or Sp1 in NSCLC patients with pemetrexed resistance (line 293 of the manuscript), should also perform NSCLC xenografts experiments in NOD/SCID using Mithramycin, otherwise, authors only can conclude the potential clinical use of the inhibitors of BMI1.
- Explain in more detail lines 258-259 of the Discussion section “In addition, the pharmacodynamics ………. chemoresistance”)
Author Response
- It will be interesting to determine whether treatment with pemetrexed induces the expression of BMI1 or contrary it selects those tumor A549 cells that express increased levels of BMI1, previous to pemetrexed treatment. Therefore, authors can quantify BMI1, TS expression +/- pemetrexed, 1.5 uM, treatment at short time and check possible induction of the expression of these proteins.
Responses:
We thank the comments from the reviewer. After treatment of pemetrexed at 1.5 mM for 72 hours in A549 cells, the expression of BMI1 or Sp1 was downregulated, while it did not influence the expression of TS. These results may reflect that the protective role of high BMI1/Sp1 in NSCLC cells when response to pemetrexed treatment.
- Figure 5B. Panel showing TS expression is missing.
Responses:
The TS expression in BMI1 overexpressing A549 cells has been shown in Fig. 3A and the results revealed that BMI1 overexpression in A549 cells increased TS expression.
- NSCLC xenografts experiments in NOD/SCID mice do not have enough controls. Authors demonstrate that PTC-209 treatment of A400-tumors enhanced the therapeutic effect of pemetrexed. But, as understood in the section of Materials and Methods, only the group of tumors treated with PTC-209 are also treated with pemetrexed, but not the vehicle control group. To ascertain the role of increased BMI1 in the pemetrexed treatment, A400-control tumors (which supposedly express high levels of BMI1) must also be treated with pemetrexed, and then compared with A400-PTC-209/pemetrexed tumors (expressing disminished levels of BMI1).
Measure of levels of expression of BMI1, SP1 and TS in all type of A-400 tumors (with/without PTC-209, pemetrexed treatments) is needed.
Responses:
We apologize for the unclear descriptions of the groups in our animal experiments. Actually, all the A400 tumor bearing mice received pemetrexed treatment. Before pemetrexed treatment, the tumor bearing mice were divided into two groups: group 1 was the vehicle control group by injection of Cremophor/NaCl solution and the group 2 was the PTC-209 pre-treated group by injection of a dose of 20 mg/kg. All the pre-treatments were performed as 5 days per week for a total of 3 weeks.
We have revised the Fig. 7A and the legends of Fig. 7 to make it clearly.
- In the Discussion section, if authors suggest the potential clinical use of the inhibitors of BMI1 or Sp1 in NSCLC patients with pemetrexed resistance (line 293 of the manuscript), should also perform NSCLC xenografts experiments in NOD/SCID using Mithramycin, otherwise, authors only can conclude the potential clinical use of the inhibitors of BMI1.
Responses:
We agree with the comments from the reviewer. The descriptions have been revised as following: ” Our results indicate that pretreatment of PTC-209 could re-sensitize pemetrexed -resistant A400 tumors to pemetrexed (Fig. 7). It suggests the potentially clinical use of the inhibitors of BMI1 in NSCLC patients with pemetrexed resistance.”
- Explain in more detail lines 258-259 of the Discussion section “In addition, the pharmacodynamics ………. chemoresistance”)
Responses:
We have added more detail for this part of discussion as following: “For example, the expression level of ATP-binding cassette transproters or multidrug resistance-associated proteins could facilitate the remove of anti-cancer drugs from cancer cells [34-36]. In addition, the polymorphisms of the enzymes involving in xenobiotics metabolisms, such as the members of the cytochrome P450 family [37], glutathione S-transferases, or diphosphate glucuronosyltransferases, have been frequently mentioned in the influences of pharmacokinetics of anti-cancer drugs [38,39].”

Reviewer 2 Report
BMI1-SP1 pathway is well known in lung cancer progression. However, the role of BMI1 in pemetrexed resistance in NSCLC is unclear. The authors described the association between BMI1 expression level and pemetrexed sensitivity of NSCLC cells well.
Author Response
BMI1-SP1 pathway is well known in lung cancer progression. However, the role of BMI1 in pemetrexed resistance in NSCLC is unclear. The authors described the association between BMI1 expression level and pemetrexed sensitivity of NSCLC cells well.
Responses:
We thank the positive feedback from the reviewer.
Reviewer 3 Report
In this manuscript the authors study a new mechanism of resistance to permetrexed in cellular and mice models of lung cancer. In vitro experiments show that the expression of CSC related proteins Bmi1, NICD, CD44 and EMT related proteins are elevated in permetrexed resistant NSCLC cells. Then, authors focus their research in the role of Bmi1 in permetrexed resistance and the related protein SP1 and TS. Finally, authors demostrated that Bmi or SP1 inhibitors are able to sensitize permetrexed resistant cellular or mice models.
EGFR mutations are a positive factor in NSCLC. In addition, Alk, Ros1 and Ras oncogene family are involved in permetrexed resistance. It has been described that a higher status of DNA repair may contributes to permetrexed resistance, involving BER, NER, CHK1, MSH2 and Ku protein. In addition, chemoresistance is mediated by multidrug resistance ABC transporter proteins. Also, driving forces that provoques metastasis such as EMT and CSC are involved in permetrexed resistance, as is described in the article reviewed. In malignant pleural mesothelioma, the CSC marker Bmi1 is observed in permetrexed resistance.
The new contribution of this article is the description of the role of Bmi1 and SP1 in permetrexed resistance. In addition, the inhibition of these proteins could decrease permetrexed resistance in cellular and mice models.
MINOR COMMENTS
-1) In the figure 1 of the study, the authors shows the protein expression of two CSC markers, Bmi1 and CD44. I think it could be interesting to study new CSC markers in NSCLC, such as Hes1 and ALDH isoforms.
-2) The authors studied the effect of Bmi1 inhibition in cell viability.I suggest to study the influence of Bmi1 inhibition in apoptosis, cell cycle arrest and EMT related proteins.
-3) I want to ask the authors the reason to use different housekeeping proteins in their experiments, such as tubulin, β-actin and GAPDH.
-4) Authors have to include in figure legends the number of experiment replicates done.
Author Response
In this manuscript the authors study a new mechanism of resistance to permetrexed in cellular and mice models of lung cancer. In vitro experiments show that the expression of CSC related proteins Bmi1, NICD, CD44 and EMT related proteins are elevated in permetrexed resistant NSCLC cells. Then, authors focus their research in the role of Bmi1 in permetrexed resistance and the related protein SP1 and TS. Finally, authors demostrated that Bmi or SP1 inhibitors are able to sensitize permetrexed resistant cellular or mice models.
EGFR mutations are a positive factor in NSCLC. In addition, Alk, Ros1 and Ras oncogene family are involved in permetrexed resistance. It has been described that a higher status of DNA repair may contributes to permetrexed resistance, involving BER, NER, CHK1, MSH2 and Ku protein. In addition, chemoresistance is mediated by multidrug resistance ABC transporter proteins. Also, driving forces that provoques metastasis such as EMT and CSC are involved in permetrexed resistance, as is described in the article reviewed. In malignant pleural mesothelioma, the CSC marker Bmi1 is observed in permetrexed resistance.
The new contribution of this article is the description of the role of Bmi1 and SP1 in permetrexed resistance. In addition, the inhibition of these proteins could decrease permetrexed resistance in cellular and mice models.
Responses:
We thank the positive feedback from the reviewer.
MINOR COMMENTS
-1) In the figure 1 of the study, the authors shows the protein expression of two CSC markers, Bmi1 and CD44. I think it could be interesting to study new CSC markers in NSCLC, such as Hes1 and ALDH isoforms.
Responses:
We thank the suggestions from the reviewer. Because of the limited time, we only examined the expression of ALDH1A1 and ALDH1A2. After western blot analysis, the expression of ALDH1A1 or ALDH1A2 in pemetrexed A400 cells was not increased when compared to A549 cells.
These data were consistent with a previous finding from Okudela K et al. that ALDH1A1 might function as a tumor suppressor in NSCLC (Downregulation of ALDH1A1 expression in non-small cell lung carcinomas--its clinicopathologic and biological significance. Int J Clin Exp Pathol. 2013;6(1):1-12.).
The results of ALDH1A1 and ALDH1A2 protein expression were added as the Fig. 1C in this revised manuscript.
-2) The authors studied the effect of Bmi1 inhibition in cell viability.I suggest to study the influence of Bmi1 inhibition in apoptosis, cell cycle arrest and EMT related proteins.
Responses:
We thank the suggestions from the reviewer. The cell cycle analysis was performed by propidium iodide staining and the results revealed that the treatment of PTC-209 in pemetrexed resistant A400 cells induced G1 arrest in a dose-dependent manner (Fig. 4C). The expressions of EMT related proteins after knockdown of BMI1 in pemetrexed resistant A400 cells were added as Fig. 6A. The results revealed that knockdown of BMI1 in A400 cells caused the upregulation of E-cadherin and the downregulation of two transcriptional repressors of E-cadherin including Snail1 and ZEB1.
-3) I want to ask the authors the reason to use different housekeeping proteins in their experiments, such as tubulin, β-actin and GAPDH.
Responses:
These three housekeeping proteins are well-acceptable loading controls in western blot analysis for total cellular proteins. From the following data, we think these three housekeeping proteins were stable during the experimental treatment.
We believe the conclusions of this study will not be influenced with using different housekeeping proteins of b-actin/GAPDH/Tubulin.
-4) Authors have to include in figure legends the number of experiment replicates done.
Responses:
The number of experiment replicates done has been added in the figure legends.

Round 2
Reviewer 1 Report
Authors have answered adequatly the questions required.